# Personality Characteristics of Children and Adolescents with Anxiety Disorder from a Maternal Perspective: A Brief Report

**DOI:** 10.3390/bs13050404

**Published:** 2023-05-12

**Authors:** Erica da Cruz Santos, Maria-Cecilia Lopes, Fernando Ramos Asbahr, Camila Luisi Rodrigues, Fabiana Saffi, Karen Spruyt, Antonio de Padua Serafim, Cristiana Castanho de Almeida Rocca

**Affiliations:** 1Department and Institute of Psychiatry, University of Sao Paulo Medical School, Sao Paulo 05403-903, Brazil; erica.csantos@hc.fm.usp.br (E.d.C.S.); fernando.asbahr@hc.fm.usp.br (F.R.A.); camilaluisi@gmail.com (C.L.R.); fabiana.saffi@hc.fm.usp.br (F.S.); 2Child and Adolescent Affective Disorder Program (PRATA), Department and Institute of Psychiatry, University of Sao Paulo Medical School, Sao Paulo 05403-903, Brazil; maria.cecilia590@hc.fm.usp.br; 3National Institute of Medicine and Health (INSERM), NeuroDiderot, Université de Paris, 75231 Paris, France; karen.spruyt@inserm.fr; 4Department of Psychology of Learning, Development and Personality, University of Sao Paulo, Sao Paulo 05508-030, Brazil; serafim@usp.br

**Keywords:** anxiety disorder, child personality, adolescent personality, maternal behavior

## Abstract

The present study assessed the personality characteristics of children and adolescents with anxiety disorder from a maternal perspective. A total of 48 children and adolescents aged between 8 and 17 years participated in this study, which was organized as follows: a clinical group (24 children and adolescents with anxiety disorders and their respective mothers) and a control group (24 children and adolescents without psychiatric diagnosis and their mothers). The participants were submitted to the WASI, CBCL, MASC-2, and EPQ-J tests and their mothers to the SRQ-20 and PIC-2 tests. The results showed higher rates of internalizing symptoms in the clinical group. In addition, patients showed less interest in hobbies, less adherence to social organizations, impairment in social activities, and commitment to school performance compared to the control group. There was a positive correlation between the mothers’ symptoms and each of the following PIC-2 domains: somatic concern (*p* < 0.01) and psychological discomfort (*p* < 0.01). In conclusion, youths with AD showed a withdrawn and reserved personality profile, involving distrust of impulses and avoidance of interactions with peers. Furthermore, psychoemotional problems of mothers adversely influenced the perception followed by anxiety and adjustment characteristics. More studies are needed to assess the maternal personality in youths with anxiety.

## 1. Introduction

The high prevalence of psychiatric disorders in childhood and adolescence has been globally reported by epidemiological studies [1]. In the Brazilian reality, psychiatric disorders in childhood and adolescence are extremely prevalent, affecting approximately 1 in 10 Brazilian children [2]. Some disorders including insomnia, fatigue, irritability, depressive moods, concentration problems, somatic complaints, and anxiety disorders occur from childhood and might worsen throughout the life span [1,3,4,5]. In addition, mental disorders in childhood and adolescence are associated with caregiver burden, including mental disorders in women [6,7]. Regarding anxiety disorders (ADs), these have been associated with substantial disabilities and economic burdens, particularly in women [8]. For example, youths with generalized anxiety disorder have been described as regressed, with manipulative behavior, or even ritualized behavior [9]. In this context, excessive maternal control can result in the presence of anxiety symptoms in the mother, and can potentiate the child’s anxiety [10]. In view of the above, we carried out a pilot study to investigate the presence of common mental disorders in the mothers of children and adolescents with anxiety disorders and how they assess their children’s personality by comparing them with a group of mothers of children with no psychiatric history.

## 2. Material and Methods

### 2.1. Participants

This is a cross-sectional study, providing a comparison between two groups matched by gender and age: one of subjects diagnosed with anxiety disorder and a control group with subjects without psychiatric disorder. The conclusion criteria for the clinical group were individuals aged between 8 and 17, of both genders, and diagnosis of AD, according to the diagnostic guidelines of the Diagnostic and Statistical Manual of Mental Disorders DSM-5 [11]. The participants were placed in the control group if they did not present any diagnosis of a psychiatric or neurological disorder. Participation in the control group was voluntary. It was disclosed on the HC communication channels about the study and the sample was selected according to the inclusion criteria. The following diagnoses were exclusion criteria: abuse and/or dependence on psychoactive substances; transient psychotic disorders (except for obsessive compulsive disorder, due to clinical and assessment specificities); history of perinatal brain injury and history of traumatic brain injury with loss of consciousness for more than one hour; and other neurological disorders. All data used in this research were collected at the beginning of the study, after the participants signed the informed consent form, which was approved by the Ethics Committee for Analysis of Research Projects at the Hospital das Clínicas da Faculdade de Medicina da Universidade de São Paulo (HCFMUSP, Clinical Hospital, Faculty of Medicine, University of São Paulo) (CAPPesq). Data collection was in accordance with the principles of the Declaration of Helsinki. The instruments used with mothers and children were applied individually in sessions with an estimated duration of 90 min.

### 2.2. Measures

The instruments used with the mothers were as follows.

Assessment of socioeconomic status: Brazil’s economic classification criteria is composed of 12 questions referring to Convenience Items (e.g., automobile, personal computer, among others); 2 questions about Basic Services (water distribution and road surface); and 1 about the educational level of the head of the household. The sum of the points provides an estimate of average household income and economic classes are considered, with A being high, B,C intermediate and D,E low [12]. The mothers of the participants answered this questionnaire.

Personality Inventory for Children—Second Edition (PIC-2) [13]: This is an objective and multidimensional questionnaire for the assessment of children and adolescents aged between 5 and 19 years, which must be completed by parents or guardians. It consists of 275 true or false questions and the results will consider behavioral, emotional, cognitive, and interpersonal adjustment dimensions in a broad manner. The scales are divided into 3 groups, namely: 3 response validity scales (Inconsistency—INC; Dissimulation—FB; and Defensive—DF), 9 adjustment scales (Cognitive Impairment—COG; Impulsivity and Distraction—ADH; Delinquency—DLQ; Family Dysfunction—FAM; Reality Distortion—RLT; Somatic Preoccupation, SOUND; Psychological Discomfort—DIS; Social Withdrawal—WDL; Skill Deficits Social—SSK), and 21 subscales of the adjustment scales. As for the analysis of correlated measures, they were tabulated within 4 T-score ranges: ≤59T, 60–69T, 70–79T, and ≥80T. Then, the results on the other instruments were reformulated as dichotomous measures: symptoms considered to be present versus absent, descriptive statements with true or false responses, and test results. skill and performance, classified as within the normal range (default score ≥85) or in need of further evaluation (default score ≤84). Ideal cutoff scores were assigned in such a way that: (a) the frequency of each descriptor equal to or greater than that T-score was at least twice its frequency below the cut-off score, (b) the proportion of profiles equal to or greater than this approximate T-score of the base rate of the descriptor in the sample studied. In this study, the Cronbach’s Alpha consisted of 12 items and the value was α = 0.90 for the total sample. For the clinical and control group, it was = 0.86 and 0.87, respectively.

Child Behavior Checklist (CBCL) [14]: This is the most used instrument in the world to screen mental health problems in children and adolescents based on parental information. In this study, Cronbach’s Alpha value of 13 items for the total sample was α = 0.95. The internal consistence for the group was clinical, α = 0.94 and control, α = 0.93.

Self-Reporting Questionnaire (SRQ-20) [15]: The questionnaire is composed of 20 yes/no questions, of which four relate to physical symptoms and 16 to psycho-emotional disorders. In this study, the Cronbach’s Alpha was α = 0.87 for the total sample. For the clinical and control group, it was =0.79 and 0.80, respectively.

#### The Instruments Used with the Children

The Multidimensional Anxiety Scale for Children (MASC-2) [16]: This is an instrument that assesses anxiety symptoms in children and adolescents. In this study, the Cronbach’s Alpha for the clinical and control group was =0.93 and 0.91, respectively.

Intellectual level assessment: Wechsler Abbreviated Intelligence Scale [17]: i.e., the estimated full IQ calculation was used, which is derived from the score of a verbal (Vocabulary) and a non-verbal (Matrix Reasoning) subtest.

Personality Questionnaire for Children and Adolescents (EPQ-J) [18]: This instrument assesses the personality traits of children aged between 10 and 16 years, who have a minimum educational level corresponding to the third grade or who can read. Personality assessment in four dimensions: Neuroticism, Extraversion, Psychoticism, and Sincerity. In this study, Cronbach’s alpha was α = 0.81 for the total sample. The clinical group considering Neuroticism, Extraversion, Psychoticism, and Sincerity traits was =0.78, 0.64, and 0.78, respectively. For the control group it was =0.79, 0.66, and 0.78, respectively.

### 2.3. Procedure

Obtaining the history of children and adolescents from the clinical group and collection of information was carried out in three stages: (1) a study of the existing medical records in the system of the HCFMUSP Psychiatry Institute and discussions with professionals for the selection of subjects; (2) an interview with the mother, who has knowledge about the history and current behavior of the child/adolescent, followed by the application of the studied instrument (PIC-2) and scales (described below); (3) establishing rapport between the examiner and the adolescent, clarifying any doubts about the main research procedures, interviewing and applying the neuropsychological tests and scale. Regarding the control group, data collection took place in a similar way, except for the reading and screening of medical records, considering the same descriptors: in a place with conditions similar to those of the clinical group, in a room with table and chairs, without external interference.

All the data used in this research were collected at the beginning of the study, after the parents signed the Informed Consent Form (ICF) and the adolescents over 12 years old signed the Free and Informed Consent Term (FICT), according to Brazilian legislation, as approved by the HCFMUSP Ethics Committee for Research Analysis (CAPPesq).

### 2.4. Results

Appendix A shows the results of the of the analysis of the sociodemographic and clinical data of the samples of the express and control clinical groups. The comparison of sociodemographic data collected between the groups included relevant IQ, socioeconomic level, mothers’ educational level, physical symptoms, as well as any psycho-emotional disorders, internalizing, externalizing, anxiety, and depression symptoms that were significantly different, mainly in the clinical group with lower socioeconomic level (B-C: intermediate), lower mothers’ educational level (High school), and a greater presence of mothers’ psycho-emotional symptoms (See Appendix A).

#### 2.4.1. Personality Investigation

Regarding the PIC-2 scale, significant differences were found between the clinical and control for three validity indicators. The clinical group scored significantly higher for the three indicators: inconsistency or the assessment of identical or opposite answers dissimulation or the assessment of the probability of answers to statements representing an exaggeration of current adjustment problems, or the description of problems and symptoms that are not present; and defensiveness, or the assessment of infrequency or highly unlikely positive attributes, and others that represent denial of behaviors and common problems of children. It is also noteworthy that the item “Response Validity Scales” evaluates guidelines for interpretation and are used to establish the presence of minimization or denial problems (Defensive), problems of exaggeration or simulation of illness (Dissimulation), and responses that reflect inadequate attention or comprehension of PIC-2 sentences (Inconsistency). Thus, the increase in this item in the clinical group demonstrates the more significant presence of symptoms in individuals and does not affect the validity of the instrument or even the scores of other sections (See Appendix A).

The Mann–Whitney U test was used for all indicators, except for the Impulsiveness and Distraction scales, and the Social Withdrawal scale, which lost significance when compared to IQ covariates. This showed that the lower the efficiency of the individuals’ cognitive functioning, the greater their difficulty modulating emotion expression, and the greater their tendency to impulsiveness and distraction (e.g., interrupting while others are talking) and social withdrawal (e.g., keeping thoughts to themselves).

#### 2.4.2. Analysis Adjusted for Sociodemographic Indicators

The variable with a high correlation with the PIC-2 adjustment scales was the level of anxiety that mothers perceive in their children (MASC-2 result, mothers’ answers), with a significant interaction effect for all items (See Appendix A). The level of anxiety was also related to IQ, suggesting that this measure is an interference factor for understanding behavioral aspects, obtained using the PIC-2. Regarding the SRQ-20 of the assessed mothers, there was a greater effect on inconsistency, dissimulation, somatic concern, and psychological discomfort. These adjustment scales describe individuals with a tendency to present stress with somatic complaints, as well as reports of excessive sleep and chronic apathy. In addition, they may exhibit inadequate self-esteem, emotional lability, tension, nervousness, and a self-critical profile, with feelings of rejection and failure (See Appendix A). When considering the socioeconomic level of the families, the inconsistency, dissimulation, psychological discomfort, and deficit in social skills variables exhibited a greater significant interaction effect, with exception for individuals with limited social relationships and little social influence (See Appendix A). It is worth noting that the defensiveness adjustment scale, which represents the infrequency or highly unlikely positive attributes and others that represent the denial of common behaviors and problems of children, established an inverse relationship with all covariates (See Appendix A).

Lower socioeconomic and IQ levels of children and adolescents could affect the mothers’ perspectives. The greater the denial of behaviors and problems seen by mothers in their children, the lower their anxiety (for example: “I rarely need to correct or criticize my child”; “My child almost never acts selfishly”; “My child will go to bed on time, without complaining”) (See Appendix A).

All statistical analyses were performed using the SPSS version 24 statistical package. The normality of the quantitative variables was verified. The Mann–Whitney U test and Student’s *t*-test were used to analyze the quantitative variables according to their distributions. To analyze categorical variables, Fisher’s exact test was used. The significance level adopted was 5%. The statistical significance of the variables was verified using the covariation of relevant sociodemographic indicators of the study. Fisher’s linear discriminant analysis was applied to build a classifier based on sociodemographic and clinical variables, namely the PIC-2, CBCL, EPQ-J, and MASC-2 scales. The linear function considered all 42 variables. The best predictors were found based on the correlation between the variables and the discriminant function. In order to verify the quality of the model, precision was measured with the leave-one-out cross-validation method to assess the generalizability of our findings. The generalized linear model was used to analyze each variable according to the group, with covariation of the indicators. This linear model relates to the answer variable through a link function and allows the magnitude of the variance of each measurement to be a function of its predicted value.

## 3. Discussion

This study stands out for the fact that it demonstrates that youths with anxiety disorders are affected by maternal neuropsychoemotional aspects, and they presented impairments in their cognitive and emotional performance. The present study also assessed the somatic worry and psychological discomfort adjustment scale, with a higher rate of somatic symptoms in our sample. According to our data, Ref. [19] considered that children and adolescents with anxiety and/or depression had higher rates of somatic symptoms than other psychiatric disorders, characterized by headache, epigastric pain, muscle tension, sweating, and nervousness.

Children and adolescents in the clinical group had cognitive dysfunction in the reality distortion scale, being more emotionally labile, more dependent on adults, and having deficits in communication skills when compared to the control group. The affective and emotional characteristics in the clinical group were depressed mood, the presence of sadness and unhappiness, associated sleep complaints, and nightmares. They also showed signs of psychological discomfort with the presence of worries and fears. These disturbances can also be added to somatic concerns and externalizing symptoms, which may reflect feelings of anger and anxiety, behaviors of extreme concern, or pathological hypersensitivity to external stimulation. Our results corroborate other studies which showed that anxious children have negative emotional hyperreactivity, operationalized as a high frequency of emotional activation and the intensity of this response [20]. These authors also showed that anxious subjects rated a higher proportion of stimuli as negative and threatening compared to non-anxious controls.

Furthermore, we demonstrate that children and adolescents with anxiety showed higher levels of somatic concern with the presence of easy tiredness, headaches, and back pain. Another study showed that youths with anxiety disorder complained of emotional and behavioral difficulties, described as aggressiveness, anxiety, social isolation, depression, frequent crying, dependence, immaturity, fear, aggressiveness, and violation of rules [21]. Moreover, according to maternal complaints, our sample was more distracted and impulsive, also more argumentative (see Appendix A). 

It is worth noting that, in the standardized PIC-2 sample, the results showed an incidence of 26% > 70 T-score in children diagnosed with Anxiety Disorder, which corroborates the findings of this research.

Intellectual resourcefulness also impacted the expressiveness of social skills in youths. It is worth emphasizing that IQ is a variable that can have an impact on the execution of the scales, so we thought it was sensible o control it for the analysis of the constructions. Our results showed changes in the social withdrawal scale, indicating social discomfort perceived by mothers who described their children as being more embarrassed, fearful, and uncomfortable in the face of the need to talk to other people, as well as having few friends, poor relationships with peers, and with low social skills. The social skills deficits adjustment scale described the degree of commitment that influenced the presence of problematic relationships with peers. These results suggested that children with internalizing problems had social skill dysfunction, corroborating with longitudinal studies that showed that internalization pathways were associated with more impaired social skills [22,23].

Mothers that presented more symptoms of mental disorder based on SRQ-20 perceived their children as being more emotionally labile and less competent in adapting to their environment. The more signs of mental disorder in the mothers, the more children were perceived as complaining (headaches and stomachaches), apathetic, and tired. The somatic perceptions likely become more evident for mothers who have a similar profile to their children [24,25]. The clinical group showed symptoms of anxiety identified using the self-report scale (MASC-2), the scale adjustment (PIC-2), and symptoms of anxiety (MASC-2) when the mother performed the assessment.

Interestingly, on the scale of anxiety symptoms in children and adolescents (MASC-2), when considering the self-assessment performed by children and adolescents, there was no significant difference between the groups, which makes us think that youths fail to identify many of their own symptoms. The understanding of children and adolescents regarding themselves and others, and regarding social relationships, reflects or is based on their overall level of cognitive development, such as their level of perspective-taking skills [26]. However, in the assessment of the mothers, these symptoms were seen with more intensity. There have been discussions about the influence of parenting styles in the relationship with the child when there is excessive maternal control, which may result from the mother’s anxiety, also generating anxiety in the child [10,27,28] described that low levels of parental autonomy, as well as excessive involvement, can make the child feel unable to independently navigate in age-appropriate tasks.

Our results suggest there is a progressive tendency in youths to be depressive/anxious, as assessed by Self Reporting Questionnaire-20 (SRQ-20), with symptoms such as feeling nervous, tense, or worried being commonly reported [8]. We sought to investigate arguments about the impact of economic class on people’s personal and social identities, and the results tend to explain that situational factors highly interfere in economically disadvantaged groups and that working-class individuals can benefit from educational and occupational opportunities to improve their economic conditions. With this in mind, the negative correlation found in this research is confirmed [29].

However, in addition to discussions about differences in economic and social class, multiple studies have pointed out the impact of a mother’s mental health status on children’s emotional and cognitive development. In addition, factors related to lower educational level (by the mother or both parents) also have an impact; these include antisocial behavior, smoking, psychological distress, severe depression, problems with alcohol and substance abuse, antisocial personality traits or involvement in criminal activities, family conflict, marital breakdown, or violence in the family environment. All these aspects can impact cognitive and emotional development, bringing problems to their academic and social lives [3,4,5].

## 4. Conclusions

This study showed that children and adolescents with anxiety disorders have difficulties in emotional development, and we sought to create the profile of the personality characteristics of children and adolescents from a maternal perspective. To this end, two groups were investigated, one composed of individuals with anxiety disorder and the other composed of individuals without any psychiatric conditions, for comparative purposes. They were described as individuals with the prevalence of internalizing symptoms, featuring a withdrawn and discreet profile, in which they distrust the impulses of the moment and seek calmer environments, avoiding interactions with peers. While the clinical youths maintained the presence of internalizing and externalizing symptoms within the average, children and adolescents were more talkative, sociable, and available to make friends and maintain social relationships. Furthermore, it was found that the possible psycho-emotional problems of mothers adversely influence the perception they have of their children, regarding anxiety and adjustment characteristics. Given this situation, it is recommended that, concomitantly with the therapeutic intervention of the children, the mothers also receive treatment, not only related to psychoeducation for guidance on the child’s diagnosis, or forms of management and emotional assistance for the children but also care in relation to their personal condition and motherhood.

Finally, we recommend that future studies focusing on this topic include larger samples in order to allow more robust data analysis, as well as to validate not only the children’s needs but also their mothers’.

## Data Availability

Data are available in an outpatient spreadsheet due to privacy. If this database is needed for a meta-analysis, it will be sent individually.

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
