# Peer review of "Personality Characteristics of Children and Adolescents with Anxiety Disorder from a Maternal Perspective: A Brief Report"

_behavsci, 2023, doi:10.3390/bs13050404_

Round 1
Reviewer 1 Report
This article compares the characteristics of adolescents with anxiety disorder to a control group in Brazil, along with characteristics of their mothers. This information has value and I have some comments to improve the clarity of the manuscript. I’ve provided the comments in their order as they appear in the manuscript.
1. Please include measures and interpretation of effect sizes for the mean differences between the clinical and control groups. This effect size information could also be included in the abstract.
2. In the abstract, add to the sentence, “In addition, patients showed less interest in hobbies, less adherence to social organizations, impairment in social activities, and commitment to school performance” “than the control group.”
3. In the Method section, state whether participation was voluntary or compensated. I assume voluntary.
4. In the Method section, describe how were the control group solicited? Were they visitors of the HCFMUSP Psychiatry Institute or relatives of its patients?
5. In the Measures section, report measure reliabilities (Cronbach’s alpha) where applicable.
6. In the Measures section, provide more information about the constructs measured by the materials for those unfamiliar with them. For example what sort of physical symtpoms and psycho-emotional disorders does the Self-Reporting Questionnaire measure? Describe “Personality Inventory for Children – Second Edition (PIC-2)” a little more – how many constructs does it measure? What are the constructs? Does it provide a single total score?
7. In the Results section, there’s a typo error of “express.” “Table S1 shows the results of the of the analysis of the sociodemographic and clinical data of the samples of the express and control clinical groups.”
8. In Table S1, please explain what Socioeconomic levels A, B-C, D-E correspond to. I assume A is highest, D-E is lowest?
9. The authors controlled for adolescents’ IQ in comparing the groups, but it also appears there’s big differences between the clinical and control groups in terms of SES and maternal education. Perhaps this can also be discussed in the discussion.
10. In sentences that note a significant difference between groups, also state how the groups were different – which was higher or lower on the variable? “In the comparison of socio-demographic data collected between the groups, relevant IQ, socioeconomic level, moth-ers’ educational level, physical symptoms, as well as any psycho-emotional disorders, in-ternalizing, externalizing, anxiety, and depression symptoms that were significantly different, mainly in the clinical group (See Table S1).” In addition to stating they were significantly different, describe which group was significantly higher/lower for these constructs.
11. Which variables in Table S1 were analyzed using the Student’s t-test and which using the Mann-Whitney test? There are superscripts in Table S1 (1, 2) – but these aren’t denoted in the notes – do these correspond to which variables were analyzed using Student’s t-test vs Mann-Whitney test? This information is only provided for Table S2.
12. Regarding the block of text: “Personality investigation: Regarding the PIC-2 scale, it was possible to analyze significant differences between the study groups, clinical and control for the three indicators: inconsistency or the assessment of identical or opposite answers dissimulation or the assessment of the probability of answers to statements representing an exaggeration of current adjustment problems, or the description of problems and symptoms that are not present; and defensiveness, or the assessment of infrequency or highly unlikely positive attributes, and others that represent denial of behaviors and common problems of children (See Table S2).” For clarity, rephrase it: “Regarding the PIC-2 scale, significant differences were found between the clinical and control for three validity indicators.” “The clinical group scored significantly higher on…” – this would make the results clearer.
13. The inconsistency, dissimulation, and defensiveness items are listed in Table S2 under the heading “Response Validity Scales,” so given that the clinical group scored higher (inconsistency, dissimulation) and lower (defensiveness) on these than the control group, does that have implications for the validity or interpretation of these scores? Does that affect the validity of their other scores on this test? I am unfamiliar with this scale. Should this be discussed in the discussion/limitations section?
14. Please state, justify, why, IQ was controlled for in the analyses in Table S2. Is this a common covariate to control for regarding these constructs?
15. In Table 3, add a note that the coefficients represent pearson’s correlation r, to make clearer what these coefficients are.
16. In the section “Analysis adjusted for sociodemographic indicators. The variable with a high correlation with the PIC-2 adjustment scales was the level of anxiety that mothers perceive in their children (MASC-2 result, mothers’ answers), with a significant interaction effect in all items (See table S3).” It states a “significant interaction effect” – please change to “significant relationship with all items.” This is because the use of the term interaction implies a statistical interaction effect which is tested through moderated regression, and is different than the correlation analyses the authors performed. For the same reasons, please remove the term interaction from this text: “When considering the socioeconomic level of the families, the inconsistency, dissimulation, psy-chological discomfort, and deficit in social skills variables exhibited a greater significant interaction effect, with exception for individuals with limited interactions and little social influence (See tables S1 and S3).”
Author Response
Please see the attachment.
Dear reviewer,
Thanks for the remarks.
Below is a new version of the manuscript with all changed points written in blue.
Thank you very much

Reviewer 2 Report
This is an interesting brief study in which personality traits in children and adolescents and their relationships with anxiety disorders from the maternal point of view were analyzed.
Even considering that the study is brief, that is, described in its essential points, I can comment and recommend some aspects, which follow:
1) The study is well described. All important stages are well described. The problem can be easily understood, the methods (instruments) employed, the results are well summarized, followed by a clear and concise discussion.
2) Regarding the instruments, scales, tests (varied) used with the mothers and children, I would recommend pointing out in the body of the description of each one of them the national validity and reliability coefficients (psychometric indicators) (for Brazil, certainly) and to other international populations. From my point of view, it is important to indicate how good (psychometrically speaking) the instruments are;
3) By controlling the other variables measured in the instruments, what is the role of intelligence (score, IQ) among the analyzed relationships, especially as a predictor of anxiety? It would be interesting to comment or analyze statistically.
4) I believe that the manuscript, plus the recommended psychometric indicators, can be a good contribution to the area.
Author Response

(The authors gave the same response as above.)

Round 2
Reviewer 1 Report
Thank you for addressing the reviewer comments.